# Surface Heterostructure of Aluminum with Carbon Nanotubes Obtained by Laser-Oriented Deposition

**Natalia Kamanina** [1,2,3,*]**, Konstantin Borodianskiy** [4] **and Dmitry Kvashnin** [5,6,7,*]

1   Lab for Photophysics of Media with Nanoobjects at Vavilov State Optical Institute, Kadetskaya Liniya V.O., dom 5/2, 199053 St. Petersburg, Russia
2   St. Petersburg Electrotechnical University ("LETI"), Ul.Prof.Popova, dom 5, 197376 St. Petersburg, Russia
3   Scientific Center "Kurchatov Institute"—Nuclear Physics Institute, Orlova Roscha 1, Leningradskij Region, 188300 Gatchina, Russia
4   Department of Chemical Engineering, Ariel University, Ariel 40700, Israel; konstantinb@ariel.ac.il
5   Emanuel Institute of Biochemical Physics RAS, 4 Kosigina st., 119334 Moscow, Russia
6   Pirogov Russian National Research Medical University, Ostrovitianov str. 1, 117997 Moscow, Russia
7   School of Chemistry and Technology of Polymer Materials, Plekhanov Russian University of Economics, Stremyanny Lane 36, 117997 Moscow, Russia
*   Correspondence: nvkamanina@mail.ru (N.K.); dgkvashnin@phystech.edu (D.K.); Tel.: +7-(812)-327-00-95 (N.K.)

**Abstract:** Al is one of the most widely applicable metallic materials due to its advanced properties. However, its main drawback is its strength, which is relatively low compared to ferrous alloys. This issue may be resolved using different approaches. In the present work, a heterostructure of Al substrate with a modified surface with carbon nanotubes (CNTs) was studied. This heterostructure was obtained using the laser-oriented deposition technique. The obtained results showed a slight reduction in the reflectivity of the obtained Al substrate with embedded CNTs compared to pure Al. Additionally, the obtained surface heterostructure showed enhancement in microhardness and higher hydrophobicity. Simulation of the CNT embedding process revealed that CNT penetration strongly depends on the diameter. Hence, the penetration increases when the diameter decreases.

**Keywords:** aluminum; carbon nanotubes; laser-oriented deposition technique; molecular dynamics



## 1. Introduction

It is well known that various areas of human activity include aluminum as a primary or secondary material for different applications [1–6]. Aluminum and its alloys are widely used in the automotive and aircraft industry, railway station construction, and shipbuilding, as well as in general optoelectronics and in cryogenic technology. Despite the many advantages of aluminum, the low strength significantly limits its potential applications. Many attempts have now been made to improve the mechanical properties of aluminum by creating nanostructured composites with the addition of two-dimensional materials [7–14]. Thus, the improvement of the mechanical properties of these metallic materials is an issue of high scientific interest.

The basic principles of molecular physics show that the interatomic distance in crystal structures is proportional to the lattice parameter. Aluminum has a face-centered cubic (FCC) lattice with a lattice parameter equal to $4.046 \times 10^{-8}$ cm at room temperature [15]. Therefore, it is logical to assume that the formation of strong C–C bonds, which exhibit lengths of ~0.13–0.14 nm, and high Young's modulus [16,17] on the surface of Al are beneficial for the advanced performance of the surface.

$CO_2$ lasers are used for a wide variety of applications. Hence, Liu et al. used this approach to synthesize ceramic material by sintering of alumina with boron carbide additions [18]. Chen showed a sintering process of yttria-stabilized zirconia with MgO powder using a $CO_2$ laser [19]. Laser annealing was investigated by Jo et al. The authors used

a $CO_2$ laser in the annealing of Al-doped ZnO material to reduce defects that appear during fabrication [20]. The surface heterostructure fabrication using the laser-oriented deposition technique was previously described in [21,22]. The laser-oriented deposition technique was used to improve basic parameters of the inorganic crystals of KBr and LiF with carbon nanotubes (CNTs) and showed an advanced performance of the produced surface heterostructure [23,24].

Here, the innovative laser-oriented deposition technique was applied to produce a heterostructure of CNTs embedded in aluminum substrate. The obtained heterostructure was examined for reflectivity, microhardness, and wetting, and the obtained results were compared with pure Al. Additionally, a molecular dynamic simulation was performed to illustrate the process of surface heterostructure formation.

## 2. Materials and Methods

The modification of the aluminum surface was performed with an IR $CO_2$ laser operated at a wavelength of 10.6 μm with a power of 30 W and a beam diameter of 5 mm. The system was connected to a vacuum chamber. CNTs were embedded in the material's surface at an additional electric field of 100–600 V·cm$^{-1}$ which was applied to orient the CNTs in the vertical position during the deposition process. The grid to which the voltage was applied was placed at a variable distance in front of the samples. This made it possible to change the electric field's strength and conduct directional oriented CNT deposition. It should be mentioned that this method does not require the creation of any additional conditions for heating of the substrates and the composition of the gas reagents, which favorably differs from the classical methods based on the chemical vapor deposition and physical vapor deposition processes. Thus, the laser-oriented deposition (LOD) method was realized efficiently.

The Al substrate with a diameter of 35 mm and a thickness of 5 mm used in the work is shown in Figure 1. The CNTs used in the work were single-wall carbon nanotubes (SWCNTs) type #704121, with a diameter in the range of 0.7–1.1 nm (Aldrich Co., Karlsruhe, Germany). The spectra of the nano-object-treated Al materials were obtained using the Perkin-Elmer Lambda 9 and the Furrier FSM-1202 instruments ("Nica-Garant+", Saint-Petersburg, Russia). The contact angle (CA) was tested using the OCA 15EC device (LabTech Co., Saint-Petersburg-Moscow, Russia). Contact angle values presented in this work are the average of five points on each of five tested samples.

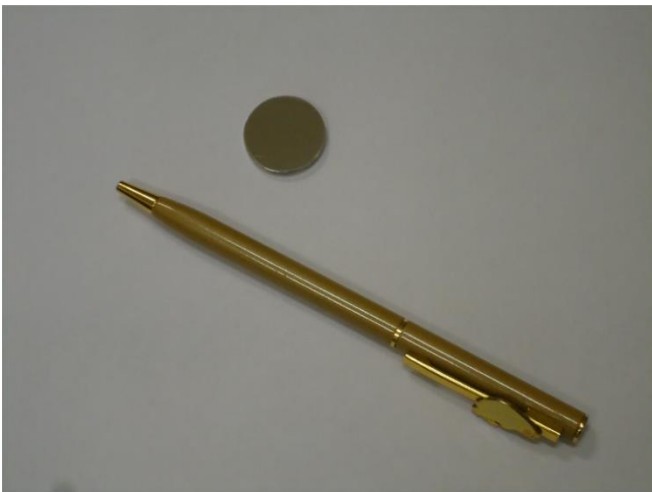

**Figure 1.** The examined Al substrate.

Simulations of CNT deposition into the Al surface were performed using classical molecular dynamics using the LAMMPS simulation package [25]. The Al(111) surface was considered as the most favorable in terms of energy. Behavior of the Al substrate at a finite

temperature was examined by means of the embedded atom method (EAM) [26] which is applicable for metal simulations.

The Al substrate–CNT interaction was described by Lennard-Jones potential according to the equation:

$$E = 4\varepsilon \left[ \left( \frac{\sigma}{r} \right)^{12} - \left( \frac{\sigma}{r} \right)^{6} \right] \tag{1}$$

where $\varepsilon = 2.63$ eV, $\sigma = 1.91$ Å. CNT behavior at the final temperature was described with a well-developed theoretical approach of the Tersoff many-body potential [27], which was successfully applied in the description of the mechanical properties of graphene membranes under defects [28]. The Al substrate included ~45,000 atoms with a total height of about 9 nm which was enough to protect CNTs from the penetration through the substrate. The molecular dynamics simulation was carried out at a constant temperature of 300 K. The total time of simulation was 60 ps with a time step of 1 fs. The acceleration rate of the CNTs varied from 100 to 600 m/s according to the experimental set up.

## 3. Results

The reflectance spectra of the pure Al and CNTs embedded in Al substrate are shown in Figure 2. Results were obtained by specular reflection at 45°.

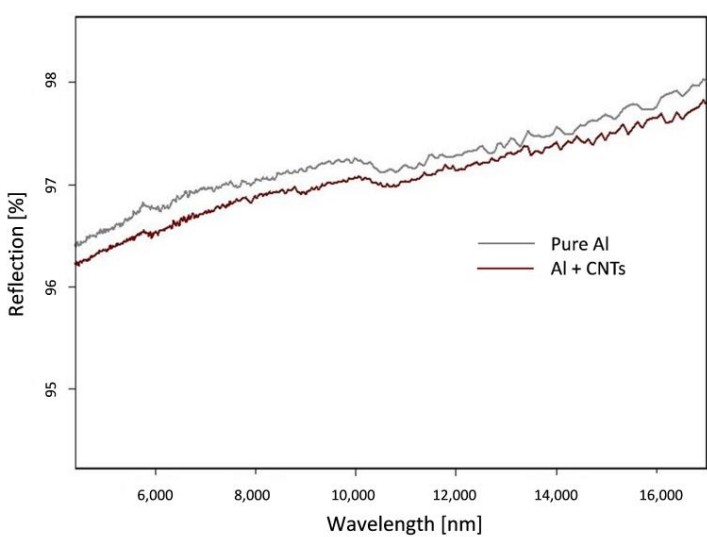

**Figure 2.** The reflectance spectra of CNTs embedded in Al substrate and pure Al.

The reflectance values at a wavelength range of 6000–16,000 nm of CNTs embedded in Al substrate showed lower values in comparison with the pure Al. This behavior may be attributed to the decrease in the Fresnel losses in the structured system. The obtained data fully correlate with the performance of a smooth aluminum surface whose reflectivity is more than 90% at a wavelength range of 900–12,000 nm, and for a wavelength of 200 nm, the reflectivity was reduced to 70%.

The wettability of the heterostructure surface was examined and compared with the surface of pure Al. The obtained data for CA are shown in Table 1. It was found that the CA of the CNTs embedded in the Al substrate is larger than that of pure Al. This behavior may be attributed to the lotus effect; it is well known that nanostructured materials usually exhibit higher hydrophobicity.

**Table 1.** Contact angle (CA) data for CNTs embedded in Al substrate and pure Al.

| Material | Contact Angle (°) | Increase (%) |
|:--------:|:-----------------:|:------------:|
| Pure Al | 108 | |
| Al + CNTs | 111 | +3 |

The microhardness values obtained are shown in Table 2. CNTs embedded in the Al substrate had microhardness of 96.3 MPa while the pure Al had a value of 90.3 MPa. This behavior may be attributed to the presence of harder CNTs on the surface of the Al.

**Table 2.** Microhardness data for CNTs embedded in Al substrate and pure Al.

| Material | Microhardness (MPa) | Increase (%) |
|:---:|:---:|:---:|
| Pure Al | 90.3 | |
| Al + CNTs | 96.3 | +7 |

The large-scale molecular dynamic simulation process was performed to understand the atomic structure behavior of CNTs embedded in the Al substrate. The simulation was performed based on CNTs with various diameters in the range of 0.64–3.35 nm which were accelerated from 100 to 600 m/s.

Figure 3 shows the plot of the penetration depth as a function of the CNT acceleration rate. It is clearly seen that the embedding of narrower CNTs led to the more local deformation of the Al surface together with their deeper penetration. Meanwhile, the atomic structure of narrower CNTs is much less sensitive to the structural distortions than wider CNTs. Here, the penetration limit of the wider CNTs is slightly over 2 nm while the penetration of the narrower CNTs reaches 5 nm. This behavior may be attributed to the decrease in curvature of the wider CNTs with a subsequent decrease in the bending modulus. Moreover, the simulation also revealed that the penetration depth of CNTs independent of its orientation towards the substrate surface. The penetration of two nanotubes of a diameter of 0.64 and 0.94 nm was also simulated at 20° related to the surface. Negligible changes in penetration depth are found for both CNTs.

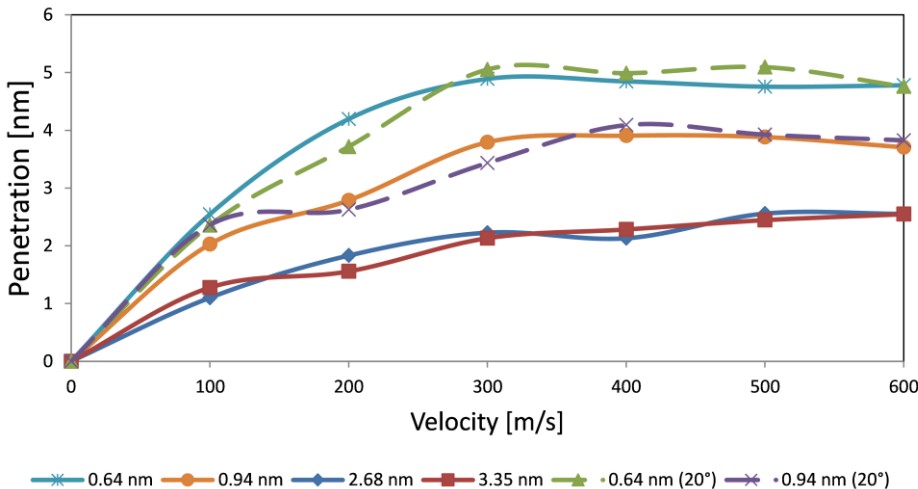

**Figure 3.** The plot of penetration depth of CNTs in Al substrate as a function of their acceleration rate.

Figure 4a illustrates a schematic presentation of a CNT with a diameter of 0.64 nm embedded in Al substrate and Figure 4b illustrates a schematic presentation of a CNT with a diameter of 2.68 nm embedded in Al substrate. The simulation was performed using an acceleration rate of the CNTs of 300 m/s. As expected, these images revealed that the narrower CNT penetrates deeper into the Al substrate than the wider one.

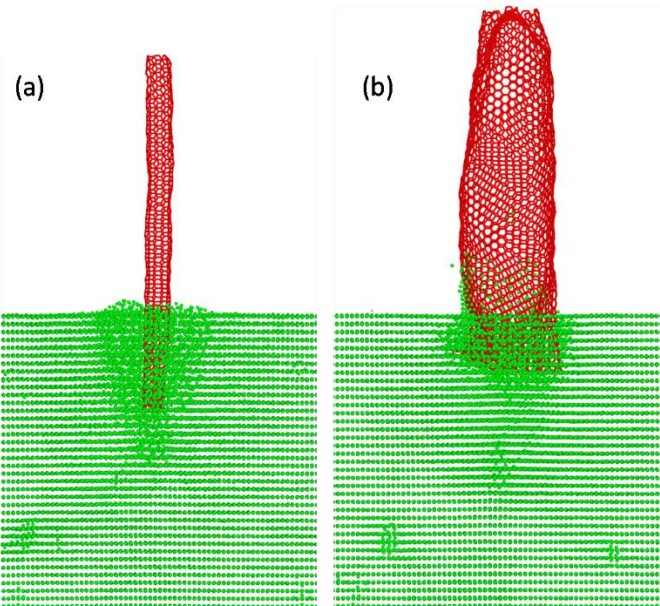

**Figure 4.** Schematic presentation of the atomic simulated structure of (**a**) CNT with a diameter of 0.64 nm embedded in Al substrate, and (**b**) CNT with a diameter of 2.68 nm embedded in Al substrate. Simulation was done using acceleration rate of 300 m/s.

### 4. Conclusions

In this work, an Al surface heterostructure was produced by the embedding of CNTs in Al substrate using the laser-oriented deposition technique to improve its mechanical and optical properties. It was shown that the reflectivity of the surface heterostructure exhibits lower values compared to pure Al. The contact angle of the surface of CNTs embedded in Al substrate increased from 108° to 111°, which is well correlated with the hydrophobicity of nanostructures according to the lotus effect. Microhardness tests also revealed a slight improvement in the surface heterostructure by 7%. Finally, a simulation of the CNT embedding process showed that narrower CNTs penetrate deeper into the substrate. Hence, the penetration of wider CNTs, with a diameter of 2.68–3.35 nm, is slightly over 2 nm while the narrower CNTs, with a diameter of 0.64 nm, penetrate over 5 nm.

It should be mentioned that some limit in size to form the coinciding conditions between the metal and CNT is related to the lattice parameter of the substrate and the diameter of the CNT. The diameter of the CNT should be less than the lattice parameter by no more than 0.1–0.2 nm. However, our previous study showed that due to the defects presence in the substrate, which usually lead to lattice deformations, such a limitation cannot be achieved. Therefore, practically, there are no restrictions on the CNT diameter.

Indeed, it should be remarked that other alternative experiments should be carried out to support the covalent bonding between the CNTs and matrix material surface atoms. This will be performed in the future and will be presented in comparison with other materials treated with the LOD technique.

**Author Contributions:** Conceptualization, N.K.; Formal analysis N.K.; Investigation, N.K.; Methodology, D.K.; Resources, D.K.; Simulation, D.K.; Visualization, K.B.; Writing—original draft preparation, N.K. and K.B. All authors have read and agreed to the published version of the manuscript.

**Funding:** The experimental results were partially obtained and analyzed via Russian National Base Project "Nanocoatings" (2012–2015). D.G.K. acknowledges the Ministry of Science and Higher Education of the Russian Federation (project No. 01201253304) for partial financial support.

**Institutional Review Board Statement:** Not applicable.

**Informed Consent Statement:** Not applicable.

**Data Availability Statement:** Publicly available data on laser-oriented CNTs deposition can be found in the patent: Kamanina, N.V.; Vasilyev, P.Y.; Studeonov, V.I. Optical Coating Based on Oriented in the Electric Field CNTs for the Optical Devises, Micro- and Nanoelectronics under the Conditions When the Interface: Solid Substrate-Coating Can Be Eliminated. RU Patent 2 405 177 C2, 23.12.2008.

**Acknowledgments:** The authors would like to thank their colleagues from the labs and institutes for the kind discussion and remarks. In particular, the authors would like to acknowledge P.Ya.Vasilyev for his help in running experiments with the laser-oriented deposition process.

**Conflicts of Interest:** The authors declare no conflict of interest.

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
