# Peer review of "Surface Heterostructure of Aluminum with Carbon Nanotubes Obtained by Laser-Oriented Deposition"

_coatings, doi:10.3390/coatings11060674_

Round 1

Reviewer 1 Report

Authors compared the penetration depth of the CNTs of several diameters. What would be the limit value of the CNT's diameter for the optimal heterostructure formation?

Try not to write a scientific article in the first person, use the neutral form "one" instead of "we".

The scanned pages with suggested corrections are enclosed.

Author Response

Reviewer #1:

Dear reviewer,

Thank you very much for your valuable remarks; we really appreciate them. The manuscript was strictly revised according to them.

Please find detailed comments to your remarks, and changes in manuscript which are marked in yellow.

Authors compared the penetration depth of the CNTs of several diameters. What would be the limit value of the CNT's diameter for the optimal heterostructure formation?

Response:

Some limit in size to form the coinciding condition between metal and CNT is related to the lattice parameter of the substrate and the diameter of the CNT. The diameter of the CNT should be less than the lattice parameter by no more than 0.1-0.2 nm. However, our previous study have showed that due to the defects presence in the substrate, which are usually led to a lattice deformations, such a limitation cannot be achieved. Therefore, practically, there are no restrictions on the CNT diameter.

This paragraph has been added in Conclusion part.

Try not to write a scientific article in the first person, use the neutral form "one" instead of "we".

Response:

Changed.

The scanned pages with suggested corrections are enclosed.

Response:

Unfortunately, we did not find scanned pages you have mentioned.

Reviewer 2 Report

 The deposition on Aluminium to increase its mechanical properties is a topical, interesting and suitable research for Coatings journal. The paper is well organized and clearly written and all such points   are merits of the the manuscript.

To increase the paper quality before publication is a need for major  revision according to the followings :

  1. A better presentation of novelty of CNT deposition with properly selected references ;the indicated references18 and 19 are patents from 2007 and 2008 and more informations have to be introduced with new references as well
  2. It is not clearly presented if the deposition was performed in the frame of this work In the conclusion is a statement  as following „In this work is Al surface heterostructure was produced by the embodiment of CNTs  in Al substrate using laser-oriented deposition technique to improve its mechanical and  optical properties”, but in the Materials and methods the description of this part of work is confusing introducing references to other paper, reference 20.                                                                                                                                                      Also the authors acknowledge Dr.P.Ya.Vasilyev, for  the deposition of the CNTs on the proposed metal in order to reveal the effects studied.  Seems to me that  deposition is not  a part of the manuscript work.
  3. The contact angle increase after CNT embedded in the Al’s surface is too small being only three and the microhardness enhancement is also only 7%.

I do suggeest more experiments

Author Response

Reviewer #2:

Dear reviewer,

Thank you very much for your valuable remarks; we really appreciate them. The manuscript was strictly revised according to them.

Please find detailed comments to your remarks, and changes in manuscript which are marked in yellow.

The deposition on Aluminium to increase its mechanical properties is a topical, interesting and suitable research for Coatings journal. The paper is well organized and clearly written and all such points   are merits of the the manuscript.

To increase the paper quality before publication is a need for major  revision according to the followings :

  1. A better presentation of novelty of CNT deposition with properly selected references ;the indicated references18 and 19 are patents from 2007 and 2008 and more informations have to be introduced with new references as well

Response:

The following was added to the text:

CO2 laser is used for wide variety of applications. Hence, Liu et al. used this approach to synthesize ceramic material by sintering of alumina with boron carbide additions [18]. Chen showed sintering process of yttria-stabilized zirconia with MgO powder using CO2 laser [19]. Laser annealing was investigated by Jo et al. Authors used CO2 laser in annealing of Al-doped ZnO material to reduce defects appeared during the fabrication [20].

The following references were added:

  • Liu, R.Z.; Chen, P.; Wu, J.M.; Chen, S.; Chen, A.N.; Chen, J.Y.; Liu, S.S.; Shi, Y.S.; Li, C.H. Effects of B4C addition on the microstructure and properties of porous alumina ceramics fabricated by direct selective laser sintering. Int. 2018, 44, 19678-19685. https://doi.org/10.1016/j.ceramint.2018.07.220.
  • Chen, F.; Wu, J.M.; Wu, H.Q.; Chen, Y.; Li, C.H.; Shi, Y.S. Microstructure and mechanical properties of 3Y-TZP dental ceramics fabricated by selective laser sintering combined with cold isostatic pressing. J. Lightweight Mater. Manuf. 2018, 1(4), 239-245. https://doi.org/10.1016/j.ijlmm.2018.09.002.
  • Jo, G.H.; Ji, J.H.; Masao, K.; Ha, J.G.; Lee, S.K.; Koh, J.H. CO2 laser annealing effects for Al-doped ZnO multilayered films. Int. 2018, 44, S211–S215. https://doi.org/10.1016/j.ceramint.2018.08.112.

  1. It is not clearly presented if the deposition was performed in the frame of this work In the conclusion is a statement  as following „In this work is Al surface heterostructure was produced by the embodiment of CNTs  in Al substrate using laser-oriented deposition technique to improve its mechanical and  optical properties”, but in the Materials and methods the description of this part of work is confusing introducing references to other paper, reference 20.                                                                                                                                                      Also the authors acknowledge Dr.P.Ya.Vasilyev, for  the deposition of the CNTs on the proposed metal in order to reveal the effects studied.  Seems to me that  deposition is not  a part of the manuscript work.

Response:

We sorry for the confusing, therefore, the sentence with the reference 20 in Materials and Methods was deleted.

In acknowledgements, the following was re-written:

Especially, authors would like to acknowledge Dr.P.Ya.Vasilyev for his help in running experiments on laser-oriented deposition process

  1. The contact angle increase after CNT embedded in the Al’s surface is too small being only three and the microhardness enhancement is also only 7%.

I do suggest more experiments

We agree that the additional study of the obtained properties have a scientific interest. However, we respectfully disagree with you that it should be presented in current manuscript, as we feel that all goals of the study have been fully achieved.

Round 2

Reviewer 2 Report

It is true that the revised version is a better paper introducing new details and references sustaining novel character, but  the main recommendations from the first review need more attention . It is still confusing  if the deposition is a novelty for this article and  for the suggestion to introduce new experiments in order to have more novelty authors answer is "we do  feel that all goals of the study have been fully achieved"

 In the absence   for a selection such as "moderate revision", my recommendation now is "accept after minor revision"

Author Response

31.05.21

Dear reviewer!

Thank you very much once again for your valuable remarks; your recommendation to explain the novelty of the LOD technique has been accepted. We have already to put some information about the scheme and marked this paragraph in green.

Moreover, we are agreeing that other alternative experiments should be made. It will be performed in future and presented for the different materials for comparison. It should be remarked that we have used this technique to treat the electro optical crystals, metals, semiconductors, some conducting layers.

The paragraphs added are here:

In the section 2:

The modification of the aluminum surface was performed with the IR СО2-laser operated at the wavelength of 10.6 mm with the power of 30 W and a beam diameter of 5 mm. The system was connected to vacuum chamber. CNTs have been embedded in the material’s surface at an additional electric field of 100-600 V´cm-1 which was applied to orient the CNTs in the vertical position during the deposition process. The grid to which the voltage was applied was placed at a variable distance in front of the samples. This made it possible to change the electric field strength and conduct directional oriented CNTs deposition. It should be mentioned that this method does not require the creation of any additional conditions for heating of the substrates and the composition of the gas reagents, which favorably differs from the classical methods based on the Chemical Vapor Deposition and Physical Vapor Deposition processes. Thus, the laser-oriented deposition (LOD) method has been realized efficiently.

In the Conclusion part:

Indeed, it should be remarked that other alternative experiments should be made to support additionally the covalent bonding between the CNTs and matrix material surface atoms. But, it will perform in the future and will present in comparison with other materials, treated with LOD technique.

 Indeed, some special conditions should be organize for the different materials, but for all of them it is very important to find the coinciding conditions between the diameter of the CNTs and the parameters of the elementary lattice.  

Thanks a lot once again!

We hope that you can accept my explanation now.

On behalf of my colleagues,

Natalia Kamanina
